# Effects of Sodium Phosphate and Sodium Nitrite on the Pitting Corrosion Process of X70 Carbon Steel in Sodium Chloride Solution

**DOI:** 10.3390/ma13235392

**Published:** 2020-11-27

**Authors:** Yongyan Zhu, Jiayi Ding, Jianli Zhang, Liang Li

**Affiliations:** 1Jiangsu Key Laboratory of Green Synthetic Chemistry for Functional Materials, School of Chemistry and Materials Science, Jiangsu Normal University, Xuzhou 221116, China; OKOKdingjiayi@163.com (J.D.); lil@jsnu.edu.cn (L.L.); 2Jiangsu Zhongneng Silicon Industry Technology Development Co., Ltd., Xuzhou 221116, China; zhangjianli@gcl-power.com

**Keywords:** corrosion inhibitor, pitting corrosion, anodic dissolution, scanning electrochemical microscope (SECM), carbon steel

## Abstract

In this paper, effects of sodium phosphate (Na_3_PO_4_) and sodium nitrite (NaNO_2_) on the pitting corrosion of X70 carbon steel in 0.10 mol/L NaCl solution were investigated by potentiodynamic polarization technique, electrochemical impedance spectroscopy (EIS) method, scanning electron microscope (SEM) and scanning electrochemical microscope (SECM). The SECM equipment was used to observe the dynamic processes of the pitting corrosion in situ. Na_3_PO_4_ or NaNO_2_ in the sodium chloride solution decreased the local anodic dissolution and increased the pitting resistance of the specimen. By analysis and comparison, it can be concluded that the inhibition effect of Na_3_PO_4_ is mainly due to the formation of a salt film, while the corrosion inhibition of NaNO_2_ is principally attributed to a protective oxide film on the electrode surface.

## 1. Introduction

When X70 carbon steel is in a corrosive environment, passive film on its surface breaks down at local points. At these points, electrodissolution takes place while the rest of the surface remains passive. Thus, pitting corrosion occurs [1,2,3,4,5,6,7,8,9].

Using inhibitors is one of the corrosion prevention methods among others. It has been found that some inorganic anions can inhibit pitting corrosion of metals or alloys in corrosive environment. The inhibition effects and mechanisms of the anions have been studied by methods of weight loss and electrochemical measurements for more than 60 years [10,11,12,13,14,15,16,17,18,19,20,21,22,23,24,25,26,27,28,29]. However, there are still many controversies about the inhibition mechanisms of the anions. Sodium phosphate (Na_3_PO_4_) is usually considered as an effective corrosion inhibitor for metals in the corrosive environments containing Cl^−^ ions. For example, Refaey et al. [10,11] found that Na_3_PO_4_ had strong inhibition effects on the pitting corrosion of mild steel in sodium chloride (NaCl) solution and in hydrochloric acid (HCl) solution. Zuo et al. [12] investigated the inhibition effects of Na_3_PO_4_ by using potentiodynamic and potentiostatic polarization measurements and proposed that Na_3_PO_4_ inhibited the pitting corrosion of 316L SS in NaCl solution by retarding both the nucleation and the propagation of metastable pits. As an inhibitor, Na_3_PO_4_ has the advantage of relatively low cost and toxicity in comparison with sodium chromate (Na_2_CrO_4_) and sodium nitrite (NaNO_2_). NaNO_2_is also considered as a good inhibitor for the pitting corrosion of metals or alloys [10,13,14,15,16,17,18]. Refaey et al. [10] found that NaNO_2_ had strong inhibition effects on the pitting corrosion of the steel samples induced by Cl^−^ ions. Fujioka et al. [16] proposed that NO_2_^−^ ions suppressed both pit nucleation and pit propagation for a passivated iron electrode in an aerated borate buffer solution containing Cl^−^ ions at pH 8.49. Although many studies concerning the inhibition effects of Na_3_PO_4_ and NaNO_2_ have been reported, the impacts of these inhibitors on the dynamic processes of pitting corrosion are rarely reported.

Scanning electrochemical microscope (SECM) has been already used to study localized corrosion, such as the pitting corrosion of X70 carbon steel in NaCl solution and so on [30,31,32]. It can provide useful information about the corrosion behavior at a microscopic level. During the pitting corrosion processes, there are more Fe^2+^ ions around the pits, so when a suitable polarization potential is applied to the probe electrode, the magnitude of the oxidation current can reflect the initiation and the propagation of the pitting corrosion [30].

In this paper, the inhibition effects of Na_3_PO_4_ and NaNO_2_ on the pitting corrosion of X70 carbon steel in 0.10 mol/L NaCl solution have been investigated by using potentiodynamic polarization technique, electrochemical impedance spectroscopy (EIS) method, scanning electron microscope (SEM)and SECM. The purpose of this study is to explore the inhibition effects of the two inhibitors on the initiation and the propagation of the pitting corrosion, and to discuss the inhibition mechanisms in depth.

## 2. Materials and Methods

A self-made four-electrode electrochemical cell was used in the SECM measurements (CHI910B electrochemical work station, Shanghai Chenhua Instruments Co., Ltd., Shanghai, China). A stationary X70 carbon steel electrode was used as the substrate electrode, whose chemical composition (in wt.%) was 0.22 C, 1.65 Mn, 0.24 Si, 0.015 S, 0.025 P and Fe balance. The carbon steel rod (Φ = 5.0 mm) was sealed by epoxy resin in a PTFE tube, leaving only one circular cross section exposed to the electrolyte. Before each measurement, the electrode surface was abraded with No. 600, No. 1200 and No. 2000 sandpaper successively, and then washed in an ultrasonic cleaner by deionized water for 3 min. A platinum plate (1 × 4 cm^2^) was used as counter electrode. A saturated calomel electrode (SCE) was used as reference electrode. If there is no special note, all the potentials in this work are referred to the SCE. During the measurements, the specimen was immersed in the solution and faced upwards. A platinum ultramicroelectrode tip (the tip electrode, d = 25 µm) moved above the specimen. The distance between the substrate and the tip was controlled to be 30 μm with the help of a charge coupled device (CCD, TCL Technology Group Co., Ltd., Huizhou, China). The tip potential was 0.65 V, which was based on the tip electrode to monitor Fe^2+^ ions as the following reaction [30]:Fe^2+^ = Fe^3+^ + e(1)

All the substrate electrodes were held at the potential of 0.250 V higher than the open circuit potential (*E*_OCP_), thus the pitting corrosion processes on the electrode surface could be observed. The scanning area was 200 × 200 μm, the increment distance was 20 μm and the increment time was 0.5 s. The surface morphology of the electrode was observed by the SEM (Model S-3400N, produced by Hitachi Company, Tokyo, Japan) after the anodic polarization at 0.250 V (vs. *E*_OCP_) in each solution.

The Tafel and EIS curves were obtained by Solartron 1287/1260 (Solartron Analytical, Hampshire, UK). The EIS curves conducted at the *E*_OCP_. The frequency range was 0.01–100,000 Hz, and the amplitude was 10 mV. The Tafel measurements were carried out positively at the scan rate of 1 mV/s from −0.030 V (vs. *E*_OCP_) until the current density reached to about 5 mA/cm^2^.

All the solutions were prepared from regents of the analytical grade and double distilled water. All the electrochemical measurements were conducted at about 25 °C.

## 3. Results

### 3.1. The Polarization Curves in the NaCl Solution without and with Na_3_PO_4_ and NaNO_2_

Figure 1 shows the Tafel curves of X70 carbon steel in 0.10 mol/L NaCl solutions without and with 0.020 mol/L Na_3_PO_4_ (B) or 0.020 mol/L NaNO_2_ (C). In curve B or C, when the potential was higher than the critical pitting potential (*E*_pit_), the current density suddenly increased steeply, denoting the rupture of the passive film and the occurrence of the pits on the electrode surface. The presence of Na_3_PO_4_ or NaNO_2_ brought a marked *E*_pit_, which indicated the pitting corrosion was inhibited to an extent by the inhibitors. Figure 2 shows the dependence of the *E*_pit_ on the concentrations of Na_3_PO_4_ or NaNO_2_. As shown in Figure 2, the higher the concentration of the inhibitor, the more positive shift of the *E*_pit_, the less favorable condition for the pit initiation.

However, the Tafel curves could only give overall and indirect information about the inhibition effects. In order to give a clear and satisfactory explanation for the inhibition mechanism, especially the comprehensive information concerning the inhibition effects on the initiation and the propagation of the pitting corrosion, the SECM measurements were carried out. The pitting corrosion of X70 carbon steel in each solution was induced by anodic polarization. In each solution containing Na_3_PO_4_ or NaNO_2_, the applied potential was a little lower than the *E*_pit_, optimal for the observation of the conversion between the metastable pits and the stable pits. During the potentiostatic polarization process, five simultaneous SECM images (The range of the X-axis is 0~200 μm, the range of the Y-axis is 0~200 μm, and the range of the Z-axis is 0~20 nA.) of the specimen had been recorded at about the 10th, 20th, 30th, 40th and 50th min, respectively. Then, the three-dimensional images were exhibited in a figure to observe the dynamic processes of the pitting corrosion. Each current peak in the images stood for a metastable or stable pitting event in the scanned area.

### 3.2. The Pitting Corrosion of X70 Carbon Steel in the NaCl Solution

In order to comparing, the SECM measurements had been firstly carried out in 0.10 mol/L NaCl solution. As shown in Figure 3A–C, the specimen was harshly attacked by the chloride ions at *E* = −0.320 V (*E*_OCP_ = −0.570 V). Three current peaks (metastable pits or stable pits) were clearly observed at the 10th min after the potential application (A); and both the number and the extent of the pits increased as time goes on (20th min, B). The surface was almost completely active after 30 min anodic polarization (C). Thus, the dynamic processes of the pitting corrosion on the electrode surface were successfully observed by the SECM measurements. Figure 4A shows the surface morphology of the specimen after 30 min anodic polarization at the potential of *E* = −0.320 V. Obviously, after the polarization, the specimen was severely corroded by the chloride ions in the solution, which is consistent with the SECM image (C) in Figure 3.

### 3.3. The Pitting Corrosion of X70 Carbon Steel in the NaCl Solution with Na_3_PO_4_

When there was Na_3_PO_4_ in the NaCl solution, the SECM images were completely different. Figure 5 shows the processes of the pitting corrosion of the specimens at *E*_S_ = −0.290 V (the *E*_OCP_ is about −0.540 V) in 0.10 mol/L NaCl solution containing various concentrations of Na_3_PO_4_. As shown in Figure 5A1–E1), two current peaks at the 10th min (A1) and three current peaks at the 20th min (B1) were observed on the surface of the specimen in the solution with 0.020 mol/L Na_3_PO_4_. Then, one current peak disappeared; the other two current peaks propagated while another two current peaks emerged (C1). And several pits initiated (D1) and developed to stable pits (E1) after 50 min anodic polarization. The current peak at the 20th min (point “a” in B1) disappeared at the 30th min, and it did not reappear anymore. The current peaks (point “b” and “c” in B1) at the 20th min developed to stable pits and did not disappear all along. One current peak appeared at the 20th min (B2) and two current peaks were observed at the 30th min (C2) in the NaCl solution containing 0.040 mol/L Na_3_PO_4_, while only one current peak occurred at the 30th min in the NaCl solution containing 0.060 mol/L Na_3_PO_4_ (C3). When the concentration of Na_3_PO_4_ increased to 0.080 mol/L, current peak did not emerge until at the 40th min (D4). These results clearly showed that as the concentration of Na_3_PO_4_ increased, the number and the extent of the pits decreased obviously, and the induced time of the pitting corrosion increased distinctly. Figure 4C–F) shows the surface morphology of the specimens after 50 min anodic polarization at E = −0.290 V in the NaCl solution containing various concentrations of Na_3_PO_4_. Obviously, the corrosion behavior shown here is consistent with that in the SECM images (Figure 5).

### 3.4. The Pitting Corrosion of X70 Carbon Steel in the NaCl Solution with NaNO_2_

Figure 6 shows the pitting processes of the specimens at *E*_S_ = −0.080 V (the *E*_OCP_ is about −0.330 V) in 0.10 mol/L NaCl solutions containing various concentrations of NaNO_2_. As shown in Figure 6, with the increase in the concentration of NaNO_2_, both the number and the extent of the pits decreased, and the induced time of the pits increased. One current peak at the 20th min (B1) and two current peaks at the 30th min (C1) were observed on the surface of the specimen in the solution with 0.020 mol/L NaNO_2_. However, the current peak did not appear until the 30th min (C2 or C3) in the NaCl solution containing 0.040 mol/L or 0.060 mol/L NaNO_2_. When the concentration of NaNO_2_ increased to 0.080 mol/L, current peak did not emerge until the 50th min (E4). The experimental results also showed that NaNO_2_ inhibited both the number and the extent of the pits, and the inhibition efficiency increased with the concentrations. Similarly, the SEM images of the specimens (Figure 7) after 50 min anodic polarization at *E* = −0.080 V in the NaCl solutions containing various concentrations of NaNO_2_ are consistent with the SECM images in Figure 6.

## 4. Discussion

The results obtained by both the SECM and SEM measurements show that Na_3_PO_4_ and NaNO_2_ are both good inhibitors for X70 carbon steel in 0.10 mol/L NaCl solution.

In this study, the SECM equipment, as an in situ technology, was used to observe the dynamic processes of the pitting corrosion, and some detailed information could be obtained. For example, the SECM results indicated that not every current peak could propagate. Point “a” in Figure 5B1, which occurred at the 20th min and disappears at the 30th min, and it did not reappear anymore. During the anodic polarization of the electrodes, some current peaks disappear; others increase after their appearance, while new current peaks might occur at any time. In the competing processes of pits initiation and surface repassivation, some current peaks “died” and others “grew”, which indicated the breakdown and the formation of the surface film on the electrode. The current peaks disappeared soon after their occurrence named “the metastable pits”, which would be repassivated or developed to stable pits. A repassivated metastable pit might still be an active site for the following metastable pit to nucleate. The initiating and the propagating of the pits around a stable pit would cause accumulated corrosion damage, that is, a larger pit on the electrode surface might be composed of several smaller pits, as shown in Figure 4C and Figure 5E1.

It is generally accepted that pitting corrosion is caused by the rupture of the surface film on the specimen. The rupture usually occurs at some unstable sites in the film, such as dislocation and so on. These sites are usually sensitive to Cl^−^ ions. The adsorbed Cl^−^ ions will displace the passive species (such as OH^−^ and H_2_O dipoles) at the sites and, in turn, accelerate the local anodic dissolution. Thus, pitting corrosion occurs.

The inhibition effects of PO_4_^3−^ ions and NO_2_^−^ ions are attributed by some investigators to their adsorption characteristics on the electrode surface. MacCafferty proposed a competitive adsorption model between PO_4_^3^^−^/NO_2_^−^ ions and Cl^−^ ions to explain the inhibition effects of the anions on the pitting corrosion [28]. The competitive adsorption of the inorganic anions with Cl^−^ ions at the active sites on the electrode surface was considered as the reason for their inhibition. NO_2_^−^ ions, PO_4_^3−^ ions and Cl^−^ ions have the similar ionic electro-mobilities, therefore the two inorganic anions and the Cl^−^ ions may migrate into the growing micro-pits by electro-migration in the similar rates. The anions competitively migrated and adsorbed on the electrode surface, thereby slowing or preventing the increase in the concentration of Cl^−^ ions on the steel surface and in the pit solution. Sakashita and Sato [24] suggested that the inorganic anions adsorbed on the steel surface possessed a strong cationic selectivity, which might repel the aggressive Cl^−^ ions and protect the steel from being destroyed.

According to Zuo et al. [12], when the pitting corrosion of 316L SS occurred in the NaCl solution, in the pit solution, hydrolysis of Na_3_PO_4_produced HPO_4_^2−^, H_2_PO_4_^−^ and H_3_PO_4_. By the hydrolysis reaction, H^+^ ions are consumed and OH^−^ ions are produced. Thus, the pH value of the pit solution increased, the repassivation of the surface occurred and the electrodissolution in the pit decreased. Therefore, they concluded that the obvious inhibition effects of Na_3_PO_4_ on the pits growth resulted from the decreased pit solution acidity and the increased material passivity.

Figure 8 shows the SECM images during the 50 min anodic polarization process at *E* = −0.300 V (*E*_OCP_ = −0.550 V) of the X70 carbon steel electrode in 0.10 mol/L NaCl + 0.010mol/L NaOH solution (pH=2), which has the similar pH value with 0.10 mol/L NaCl solutions containing various concentrations of Na_3_PO_4_. Figure 5B is the SEM image of the specimen after 50 min anodic polarization at *E* = −0.300 V in the NaCl solution containing 0.010mol/L NaOH.

When the X70 carbon steel specimen was in the NaCl solution containing NaOH, the anodic dissolution occurred according to the following reactions:Fe → Fe^2+^ + 2e(2)
Fe^2+^ + 2OH^−^ → Fe(OH)_2_(3)
Fe(OH)_2_ + 2Fe^2+^ + 2H_2_O → Fe_3_O_4_ + 6H^+^ + 2e^−^.(4)

And Fe_3_O_4_ could convert to γ-Fe_2_O_3_, which would form compact films on the electrode surface and inhibit the pitting corrosion.

Both the SECM images (Figure 5 and Figure 8) and the SEM images (Figure 4B–F) indicate that the inhibition effects of NaOH are obviously weaker than those of Na_3_PO_4_. Therefore, the inhibition mechanism of PO_4_^3−^ions is not equal to that of OH^−^ ions.

PO_4_^3^^−^ ions can form iron phosphate with Fe^2+^ ions. Iron phosphate is a poorly soluble compound. Thus, the deposition of iron phosphate is beneficial to the formation of the passive film on the electrode surface. In this case, PO_4_^3−^ ions incorporated into the passive film, forming an improved stability against Cl^−^ ions and thus retarding the corresponding destructive action. The experimental results by the electron diffraction method had shown that the protective film formed on the steel surface in the presence of PO_4_^3−^ ions was consist of a mixture of γ-Fe_2_O_3_ and FePO_3_·2H_2_O [10,12]. Therefore, the inhibition effects of Na_3_PO_4_were the results of the adsorbed PO_4_^3−^ ions on the surface of the X70 carbon steel electrode contribute to not only the decreased pit solution acidity, but also the formation of the passive film by the deposition of iron phosphate.

According to the literature [13,14,15,16,17,18] and the experimental results (Figure 6 and Figure 7), NaNO_2_ is a good anodic inhibitor, which oxidizes ferrous ions to ferric ions thereby makes the passive film more compact.

Fujioka et al. [16] had proposed that the inhibition effects of NO_2_^−^ ions might be due to the fast reduction to NH_4_^+^ during the steel dissolution reaction,
NO_2_^−^ + 6e + 8H^+^ → NH_4_^+^ + 2H_2_O.(5)

The residual oxygen on the electrode surface promoted the oxidation of steel to produce Fe_2_O_3_ [13].

On the other hand, NO_2_^−^ ions is classified as an intermediate base according to the HSAB (Hard-Soft-Acid-Basic) principle, and Fe^2+^ ions is an intermediate acid, so NO_2_^−^ ions is preferentially combined with Fe^2+^ ions, which is the direct product of the electrochemical corrosion. And NO_2_^−^ ions can oxidize Fe^2+^ ions to Fe^3+^ ions, a hard acid. Fe^3+^ ion tends to be combined with O^2−^ or OH^−^, a hard base, and thus a ferric compound is formed on the electrode surface. Then, the defects in the passive film can be repaired by the precipitation of the stable ferric compound. Therefore, NO_2_^−^ ions suppress both the nucleation and the propagation of the pitting corrosion.

In a conclusion, it can be concluded that the inhibition effect of PO_4_^3−^ ions is mainly due to the formation of the salt film, while the corrosion inhibition of NO_2_^−^ ions owes to the oxide film formed on the electrode surface. To verify the above conclusion further, the co-effects of the two anions on the corrosion behavior of X70 carbon steel have been investigated.

Firstly, Na_3_PO_4_ (c_a_ = 0.040 mol/L) or NaNO_2_ (c_b_ = 0.040 mol/L) or the two inhibitors (c_a_ = 0.030 mol/L, c_b_ = 0.010 mol/L; c_a_ = 0.020 mol/L, c_b_ = 0.020 mol/L; c_a_ = 0.010 mol/L, c_b_ = 0.030 mol/L) were added into the NaCl solution, but the total concentration (0.040 mol/L) of the inhibitor/inhibitors remained unchanged, and then the pitting resistances of the electrodes had been compared by the SECM measurements and the EIS. Figure 9 shows the SECM images of X70 carbon steel polarized at −0.150 V (*E*_OCP_ = −0.400 V) in various solutions. According to the SECM images, the pitting resistance of the specimen decreased when there were both of the anions in the same solution. The minimum pitting resistance was observed when the concentration of PO_4_^3^^−^ions was equal to that of NO_2_^−^ ions. Figure 10 is the Nyquist plots of the EIS for X70 carbon steel in the solutions. The appropriate equivalent circuit is shown in Figure 11. The inhibition efficiency (P%) is calculated according to the following formula:$$P\%\;=\;\frac{R_{\mathrm{ct}}-R_{\mathrm{ct}}^{0}}{R_{\mathrm{ct}}}\times 100$$

*R*_ct_ and $R_{\mathrm{ct}}^{0}$ are the charge transfer resistances for the electrodes in 0.10 mol/L NaCl solution with and without inhibitor(s) respectively. The data of *R*_ct_ and P% are fitted in Table 1. It can be seen that *R*_ct_ and P% decreased when there were both NO_2_^−^ ions and PO_4_^3−^ ions in the same solution. The minimum *R*_ct_ and P% were observed when the concentration of PO_4_^3−^ ions was equal to that of NO_2_^−^ ions.

Secondly, the X70 carbon steel electrode was polarized in α mol/L Na_3_PO_4_ + β mol/L NaNO_2_ mixed solution respectively (α = 0.040, β =0; α = 0.030, β = 0.010; α = 0.020, β = 0.020; α = 0.010, β = 0.030; α = 0, β = 0.040) at the potential of *E* = 0.500 V for 10 min, then the electrode was transferred to 0.10 mol/L NaCl solution, and then the EIS measurements had been carried out. As shown in Figure 12, the pitting impedance decreased for the electrodes polarized in the mixed solutions, and when the concentrations of the two anions were equal, the lowest pitting impedance had been observed0.10 mol/L NaCl solution. In each plot, the real part at low frequency contracts, which showed a compact film formed on the surface of the X70 carbon steel specimen.

If there were both NO_2_^−^ ions and PO_4_^3−^ ions in the same solution, NO_2_^−^ ions might displace PO_4_^3−^ ions on the surface of the electrode. Therefore, the pitting resistance of the specimen decreased when there were both of the two anions in the same solution for the anions compete against each other to adsorb on the electrode, that is, the salt film would prevent the formation of the oxide film, and vice versa. When there was only one type of film, the film was stable, and when two different types of film co-existed at the same time, the interface between the two films was very active and vulnerable to the corrosive media. Moreover, under the same concentration, the interface was the least stable, and the lowest inhibition efficiency was observed.

## 5. Conclusions

The inhibition effects of Na_3_PO_4_ and NaNO_2_ on the pitting corrosion processes of X70 carbon steel in 0.10 mol/L NaCl solution have been investigated.

To study the inhibition effects, SECM, which has high spatial resolution, has demonstrated its advantage over the traditional electrochemical techniques (EIS, Tafel curves and SEM and so on) by obtaining the transient current distribution images on the electrode surface. During the pitting corrosion, using SECM, the dynamic processes of the dissolution and the repair of the passive film on the electrode surface, the transformation between the metastable pits and the stable pits, the distribution of the corrosion products, the effects of the inhibitors on the dynamic processes and more information concerning the inhibition mechanisms can be obtained.

In this study, the inhibition effects and mechanisms of PO_4_^3−^ ions and NO_2_^−^ ions proposed by the reporting papers are verified by the SECM measurements. The inhibition efficiency of the two anions increases with their concentrations. The inhibition effect of PO_4_^3^^−^ ions is due to their competitive adsorption with Cl^−^ ions at active surface sites, the decrease of the pits solution acidity and the formation of the salt film by the deposition of the iron phosphate. However, the inhibition behavior of NO_2_^−^ ions is resulted from their competitive adsorption with Cl^−^ ions and their strong oxidizing abilities.

## Figures and Tables

**Figure 1 materials-13-05392-f001:**
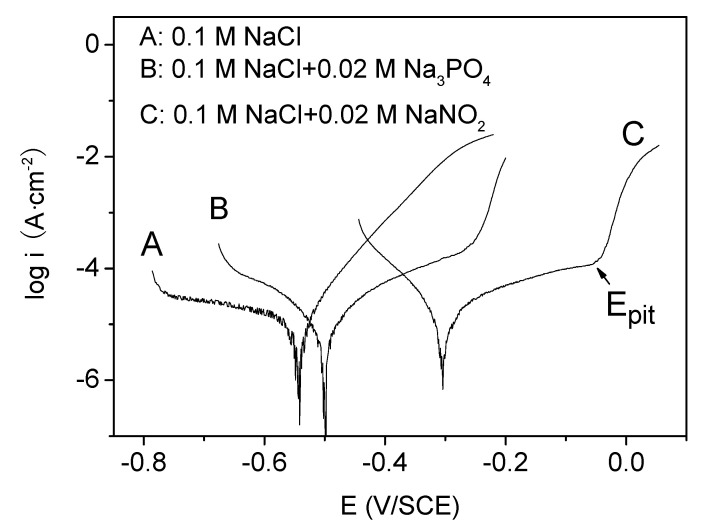
The Tafel curves of X70 carbon steel in various solutions (A) 0.10 mol/L NaCl; (B) 0.10 mol/L NaCl + 0.020 mol/L Na_3_PO_4_; (C) 0.10 mol/L NaCl + 0.020 mol/L NaNO_2_.

**Figure 2 materials-13-05392-f002:**
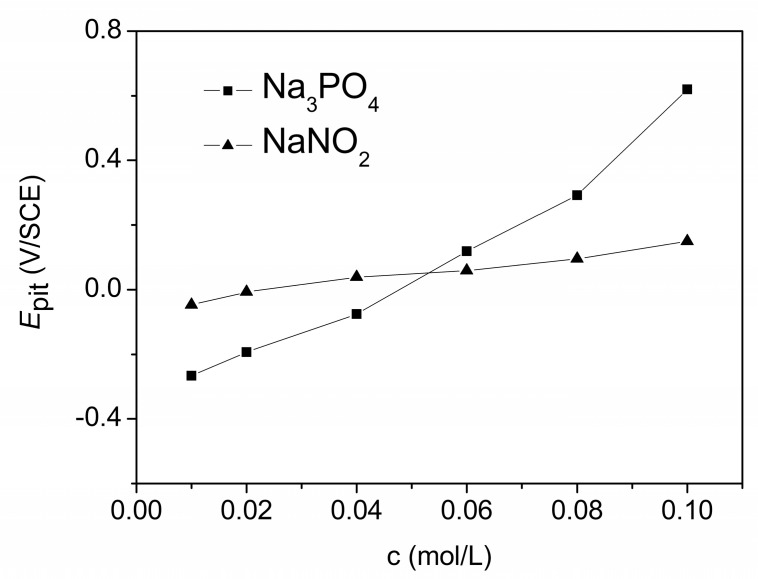
The relationship between the critical pitting potential (*E*_pit_) and the concentrations of Na_3_PO_4_ or NaNO_2_(c).

**Figure 3 materials-13-05392-f003:**
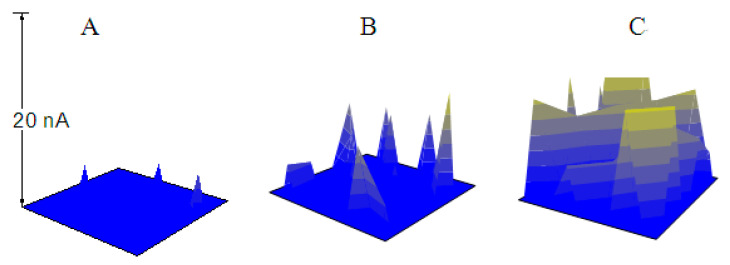
The SECM images of the X70 carbon steel electrode polarized at −0.320 V (*E*_OCP_ = −0.570 V) in 0.10 mol/L NaCl solution (**A**) 10 min; (**B**) 20 min; (**C**) 30 min.

**Figure 4 materials-13-05392-f004:**
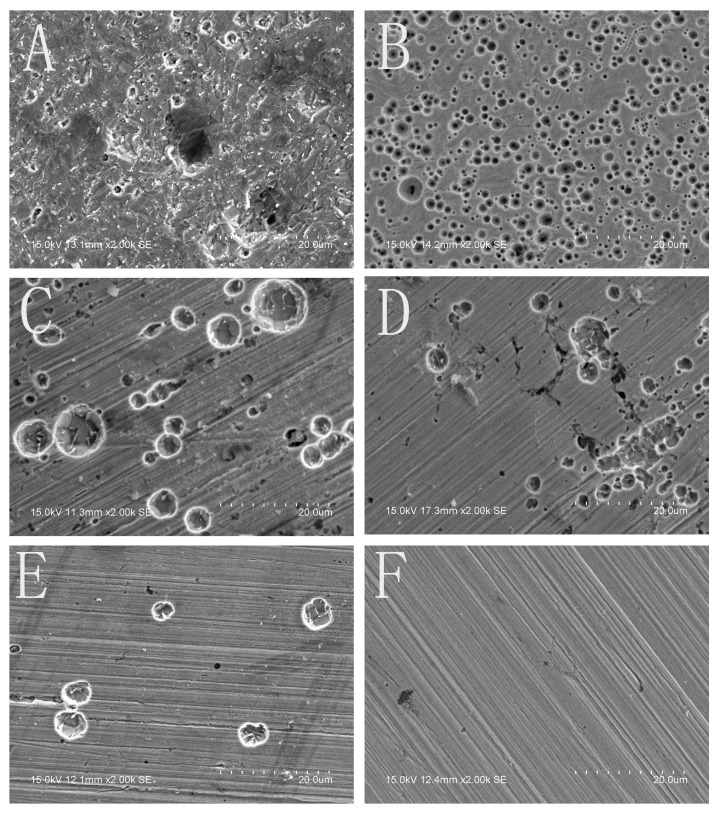
The surface morphology of the X70 carbon steel electrodes after the polarization in various solutions (**A**) after 30 min polarization in 0.10 mol/L NaCl solution at −0.320 V (*E*_OCP_ = −0.570 V); (**B**) after 50 min polarization in (0.10 mol/L NaCl + 0.010 mol/L NaOH) solution at −0.300 V (*E*_OCP_ = −0.550 V); (**C**–**F**) after 50 min polarization in (0.10 mol/L NaCl + x mol/L Na_3_PO_4_) solutions at −0.290 V (*E*_OCP_ = −0.540 V) (**C**: x = 0.020; **D**: x = 0.040; **E**: x = 0.060; **F**: x = 0.080).

**Figure 5 materials-13-05392-f005:**
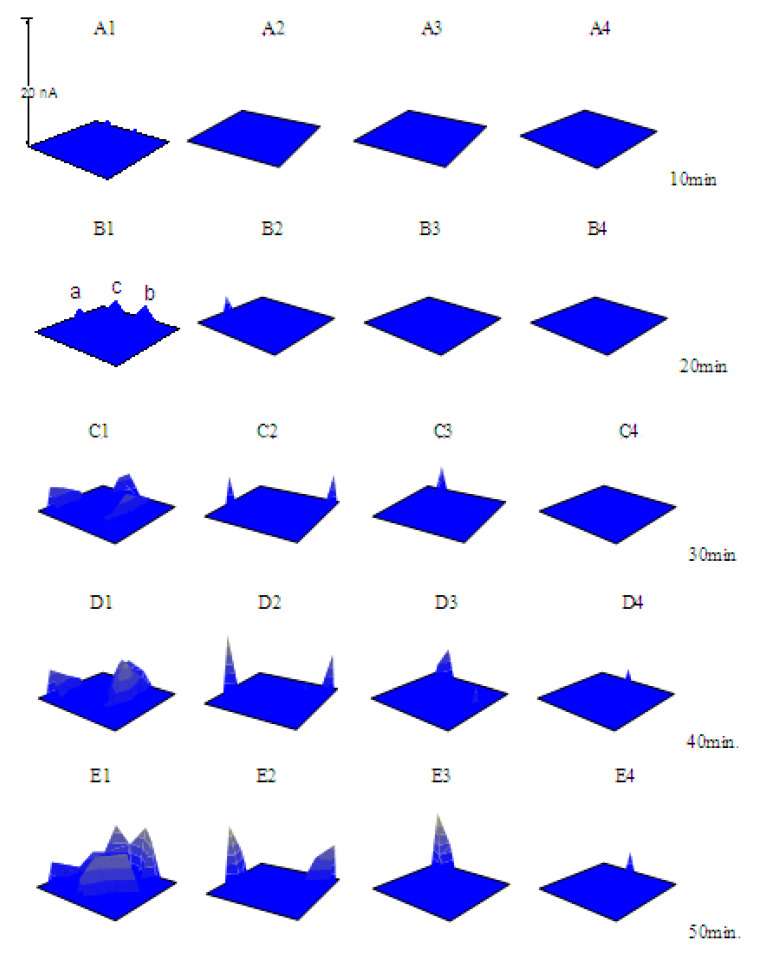
The scanning electrochemical microscope (SECM) images of the X70 carbon steel electrodes polarized at −0.290 V (*E_OCP_* = −0.540 V) in 0.10 mol/L NaCl + x mol/L Na_3_PO_4_ solutions (**A1**–**E1**) x = 0.020; (**A2**–**E2**) x = 0.040; (**A3**–**E3**) x = 0.060; (**A4**–**E4**) x = 0.080.

**Figure 6 materials-13-05392-f006:**
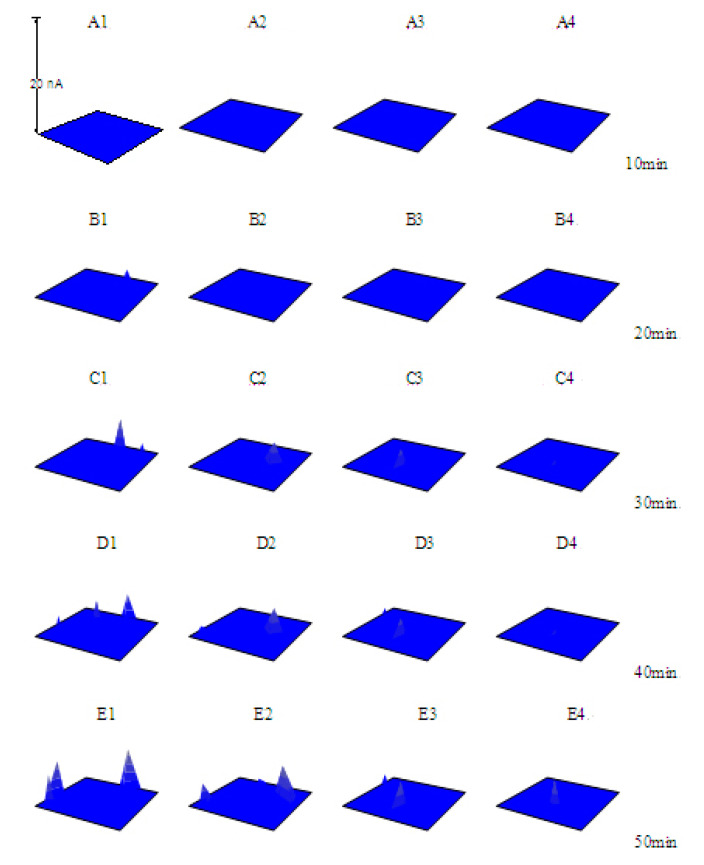
The SECM images of the X70 carbon steel electrodes polarized at −0.080 V (*E*_OCP_ = −0.330 V) in 0.10 mol/L NaCl + x mol/L NaNO_2_ solutions (**A1**–**E1**) x = 0.020; (**A2**–**E2**) x = 0.040; (**A3**–**E3**) x = 0.060; (**A4**–**E4**) x = 0.080.

**Figure 7 materials-13-05392-f007:**
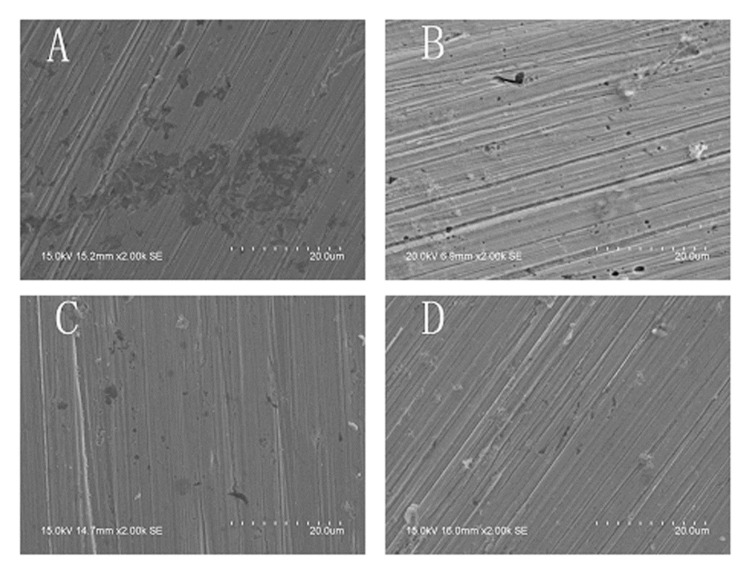
The surface morphology of the X70 carbon steel electrodes after 50 min polarization at −0.080 V (*E*_OCP_ = −0.330 V) in 0.10 mol/L NaCl + x mol/L NaNO_2_ solutions (**A**) x = 0.020; (**B**) x = 0.040; (**C**) x = 0.060; (**D**) x = 0.080.

**Figure 8 materials-13-05392-f008:**
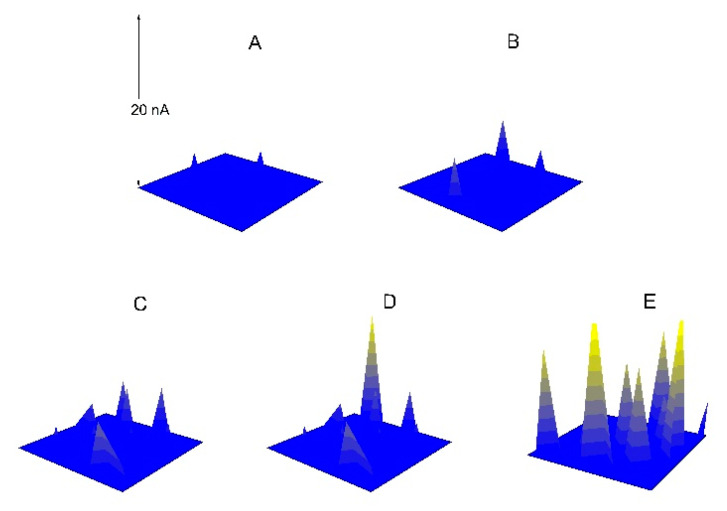
The SECM images of the X70 carbon steel electrode polarized at −0.300 V (*E_OCP_* = −0.550 V) in 0.10 mol/L NaCl + 0.010 mol/L NaOH solution (**A**) 10 min; (**B**) 20 min; (**C**) 30 min; (**D**) 40 min; (**E**) 50 min.

**Figure 9 materials-13-05392-f009:**
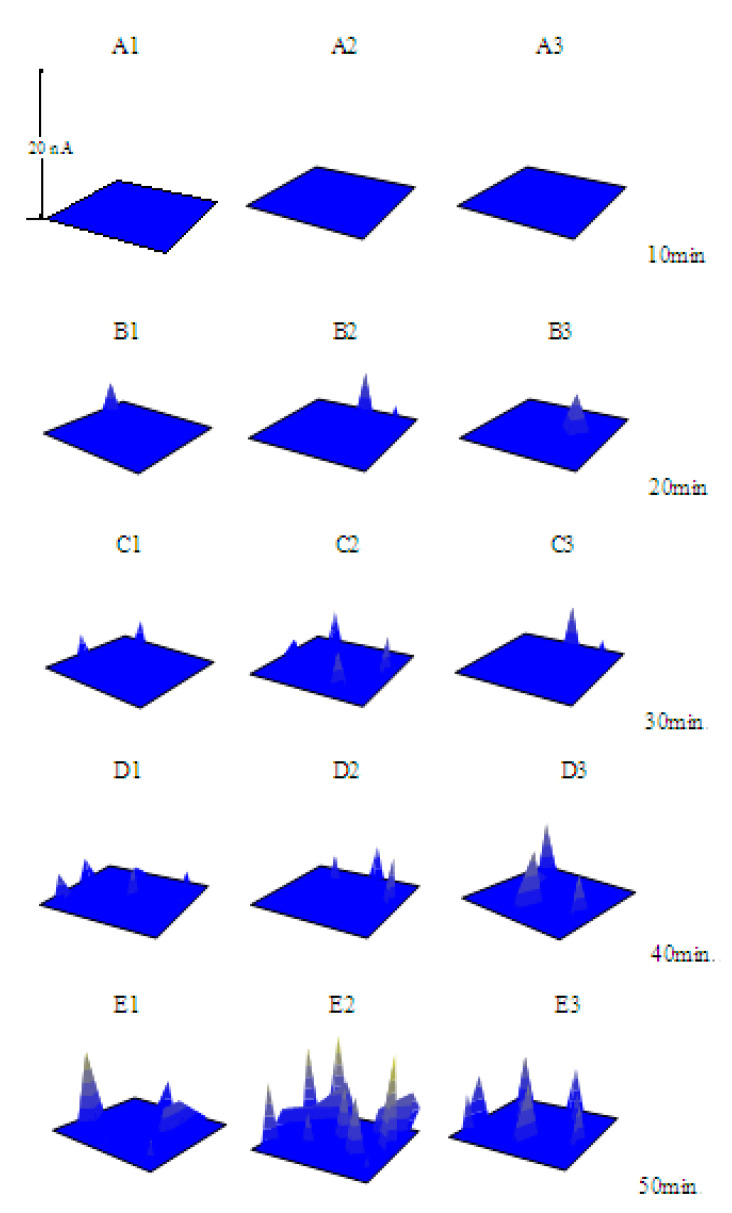
The SECM images of the X70 carbon steel electrodes polarized at −0.150 V (*E_OCP_* = −0.400 V) in 0.10 mol/L NaCl + c_a_ mol/L Na_3_PO_4_ + c_b_ mol/L NaNO_2_ solutions (**A1**–**E1**) c_a_ = 0.030, c_b_ = 0.010; (**A2**–**E2**) c_a_ = 0.020, c_b_ = 0.020; (**A3**–**E3**) c_a_ = 0.010, c_b_ = 0.030.

**Figure 10 materials-13-05392-f010:**
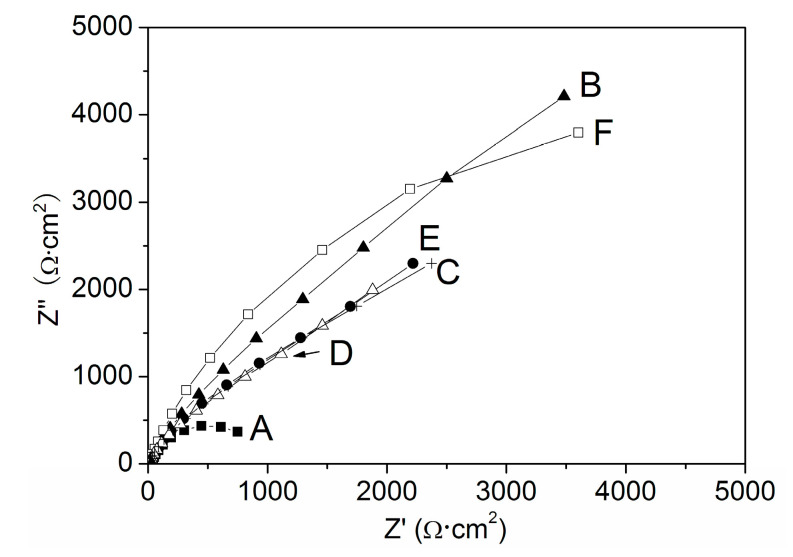
The Nyquist plots at the *E*_OCP_ for X70 carbon steel electrodes in 0.10 mol/L NaCl + c_a_ mol/L Na_3_PO_4_ + c_b_ mol/L NaNO_2_ solutions (A) c_a_ = 0, c_b_ =0; (B) c_a_ = 0.040, c_b_ =0; (C) c_a_ = 0.030, c_b_ = 0.010; (D) c_a_ = 0.020, c_b_ = 0.020; (E) c_a_ = 0.010, c_b_ = 0.030; (F) c_a_ = 0, c_b_ = 0.040.

**Figure 11 materials-13-05392-f011:**
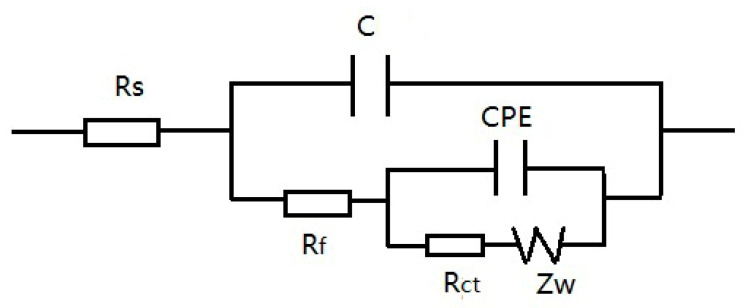
The equivalent circuit of the X70 carbon steel in 0.10 mol/L NaCl solution with Na_3_PO_4_ and/or NaNO_2_inhibitor(s). Rs: solution resistance; Rf: film resistance; Rct: charge transfer resistance; C: double layer capacitance; CPE: constant phase element; Zw: Warburg impedance.

**Figure 12 materials-13-05392-f012:**
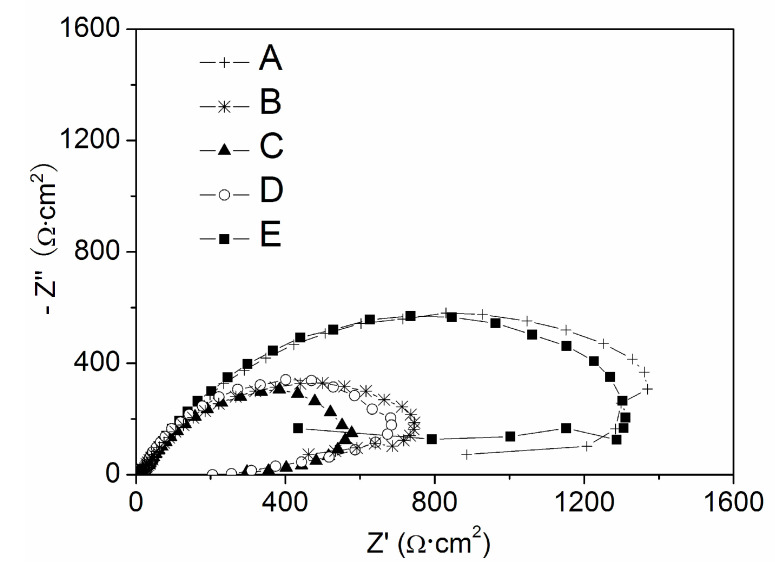
The Nyquist plots in 0.10 mol/L NaCl solutions at the *E*_OCP_ of X70 carbon steel electrodes after 10 min anodic polarization at 0.500 V in α mol/L Na_3_PO_4_ + β mol/L NaNO_2_solutions (A) α = 0.040, β = 0; (B) α = 0.030, β = 0.010; (C) α = 0.020, β = 0.020; (D) α = 0.010, β = 0.030; (E) α = 0, β = 0.040.

**Table 1 materials-13-05392-t001:** Effects of sodium phosphate and/or sodium nitrite on the *R*_ct_ and P% of X70 carbon steel in 0.10 mol/L NaCl + c_a_ mol/L Na_3_PO_4_ + c_b_ mol/L NaNO_2_ solutions((A) c_a_ = 0, c_b_ = 0; (B) c_a_ = 0.040, c_b_ = 0; (C) c_a_ = 0.030, c_b_ = 0.010; (D) c_a_ = 0.020, c_b_ = 0.020; (E) c_a_ = 0.010, c_b_ = 0.030; (F) c_a_ = 0, c_b_ = 0.040).

Solution	A	B	C	D	E	F
*R*_ct_/Ω·cm^2^	1064	3316	2172	1784	2109	3408
P%	--	67.91	51.01	40.36	49.55	68.78

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
