# Peer review of "Effects of Sodium Phosphate and Sodium Nitrite on the Pitting Corrosion Process of X70 Carbon Steel in Sodium Chloride Solution"

_materials, 2020, doi:10.3390/ma13235392_

Round 1

Reviewer 1 Report

Page 1. Line 29. Prezaazzsvention – unknown word. Did the authors mean prevention?

What is the justification for the selected concentration of inhibitors in the solution?

Part 3.2, line 125, 130. First there is a link to figure 3, and then to figure 5. It is necessary to carefully number the figures in accordance with the references to them in the text.

Part 3.2, line 130-133. In SEM image (fig 5A) readers see the same area as in fig 3C? Is it possible to explore the same surface area using SEM and SECM?

What is the rate of surface scanning when getting the SECM images?

Part 3.3, line 146. The text refers to points a, b, and c in figure 4,B1. However, they are not visible at all. You need to mark them more clearly.

Figure 4. It is logical that when the inhibitor concentration increases and the polarization potential is constant, the pitting initiation time increases (as you have shown in figure 2). It would be more useful to look at the behavior of the surface near the pitting potential for each of the inhibitor concentrations. Will the electrode behave the same in all these cases?

Line 230-234. Specify the pH value of the solution used.

Fig 9. At what potential were the presented images obtained?

Figure 10 shows an equivalent scheme for the resulting hodographs. What is the point of it if the values of its parameters are not represented? Also a question: if the hodographs shown in figure 10 are taken at frequencies of 100 kHz to 0.01 mHz, then perhaps the scheme is completely different and there is a Warburg impedance? Accordingly, the interpretation of the obtained dependencies will be somewhat different.

Figure 11 shows that the surface of the electrode is activated after 10 minutes of exposure at a potential of 0.5 V and is dissolving. We can't talk about any film based on these data. We can say that the corrosion rate in a mixture of inhibitors is higher than in mono solutions.

Methods that provide information about local processes are very interesting. However, their interpretation requires accuracy. The paper provides only a qualitative description of the obtained SECM images. No quantitative analysis was performed. The same can be said about the results of impedance measurements. It is necessary to give frequency dependencies and parameters of the equivalent circuit.

Reviewer 2 Report

Review comments

Manuscript Ref: Materials-984858

Manuscript Title: Effects of Sodium Phosphate and Sodium Nitrite on the Pitting Corrosion Process of X70 Carbon Steel in Sodium Chloride Solution

Zhu et al are using Sodium Phosphate and Sodium Nitrite to inhibit corrosion of X70 pipeline steel in NaCl. This study shows strong material component and but will not open new research directions in chemistry and/or other fields in area of corrosion. (a) This manuscript has been thoroughly read through and the general English language structure of this manuscript needs to significantly improve by the authors. (b) Since there are already numerous published papers reporting the effects of sodium phosphate and sodium nitrite in the corrosion of industrial metal substrates, it is my believe that this study lacks novelty and should not be accepted for publication in Materials.

Recommendation: Reject. Not acceptable in the current form in line with Materials guidelines; lacks novelty.

Reviewer

Reviewer 3 Report

The title of the work talks about the effect of two corrosion inhibitor substances on pitting corrosion of X70 steel. In fact, the presence of these inhibitors leads to the formation of pittng corrosion. The potentiodynamic curve for sodium chloride without the addition of inhibitors shows general corrosion with diffusion control.

The tested corrosion inhibitors belong to the group of anodic inhibitors and it is expected that in the presence of chloride ions they will be able to pitting corrosion. The authors proved such an effect of inhibitors by conducting polarization / potentiodynamic studies.  

An interesting part of the work is the use of the SECAM method to test the initiation of pitting corrosion.

Impedance studies show the differentiation in the effectiveness of the inhibitors and mixtures proposed by the authors. The authors proposed an electrical substitute diagram for EIS tests, but they do not include any calculation results. They could appear along with the error rate. It would be a natural verification of the quality of the substitute circuit selection.

The results shown in Figure 11 suggest that the measurements were performed before the test system became stable. This is indicated by measurement points in the low frequency range.

Rather, the conclusions are a collection of knowledge that is known from the literature dealing with the evaluation and testing of corrosion inhibitors. It is very difficult to assess this part from the qualitative point of view.

Round 2

Reviewer 2 Report

Review comments

Manuscript Ref: Materials-984858

Manuscript Title: Effects of Sodium Phosphate and Sodium Nitrite on the Pitting Corrosion Process of X70 Carbon Steel in Sodium Chloride Solution

Authors are using studying chloride-induced pipeline steel corrosion in the presence of Sodium Phosphate and Sodium Nitrite. The use of SECM in investigating the dynamics of pitting process distinguishes the present study from variant earlier reports. It shows strong analytical material component and will opens new research directions in corrosion and/or other fields in area of material degradation. The revised manuscript has been read through and the following corrections/queries need to be fixed in order to improve the quality of this manuscript.

  • Authors should distinctly ascribe the origin of inhibition of local anodic dissolution in the sodium chloride solution to Na3PO4 or NaNO2 or both. Since there is corrosion inhibition between both compounds, it would be great that they also point their optimum inhibition concentration.
  • For all SECM micrographs (3,5,6,8 and 9), authors are advised to uphold all axes information (x,y and z) and their labels in order to maintain clarity (as show in Figure 2 of this article: Corrosion Science 80 (2014) 511–516).

Reviewer

Reviewer 3 Report

The authors agree that the corrosion process takes place with diffusion control. However, the evaluation of the results is performed using a typical electric Randles equivalent circuit. It would be possible to convert the obtained results using a system containing some representation of diffusion.
